

# Plantar support adaptations in healthy subjects after eight weeks of barefoot running training

Celso Sánchez-Ramírez[1] and Luis M. Alegre[2]

[1] Escuela de Ciencias de la Actividad Física/Facultad de Ciencias Médicas, Universidad de Santiago de Chile, Santiago, Región Metropolitana, Chile
[2] Grupo de Investigación GENUD, CIBER de Fragilidad y Envejecimiento Saludable, Universidad de Castilla La Mancha, Toledo, Toledo, Spain

Corresponding author
Celso Sánchez-Ramírez,
celso.sanchez@usach.cl

## ABSTRACT

**Background:** Although the studies of barefoot running have intensified, it is still missing longitudinal work analyzing the effects of barefoot running on the phases of plantar support. The objective of this research was to analyze the modifications undergone by the Total Foot Contact (TFC) phase and its Flat Foot Phase (FFP) in subjects beginning the practice of barefoot running, in its acute and chronic effects.

**Methods:** A total of 28 subjects were divided into the Barefoot Group (BFGr) ($n = 16$) and the Shod Group (SHGr) ($n = 12$), evaluated before (Baseline) and after running for 20 min at 3.05 m·s$^{-1}$ (Post 20 min Running), and at the end of a running training protocol with an 8-week long progressive volume (Post-8-week Training). The dynamic plantar support was measured with a baropodoscope. The duration of TFC (ms), the moment at which the FFP occurred, the maximum surface of TFC (MSTFC) (cm$^2$), the FFP surface (SFFP) (cm$^2$), the peak pressure of TFC (PP°TFC) (kg·cm$^{-2}$), and the peak pressure of FFP (PP°FFP) (kg·cm$^{-2}$) were recorded. The $3 \times 2$ ANOVA analysis was made to determine the effects and interactions that the condition produced (Shod/Barefoot), and the time factor (Baseline/ Post 20 min Running/Post-8-week Training).

**Results:** The condition factor caused more significant effects than the time factor in all the variables. Duration of TFC in BFGr showed significant differences between the Baseline and Post-8-week Training ($p = 0.000$) and between Post-20-min Running and Post-8-week Training ($p = 0.000$), with an increasing trend. In the moment at which the FFP occurred a significant increase ($p = 0.029$) increase was found in Post-20 min Running (48.5%) compared to the Baseline (42.9%). In MSTFC, BFGr showed in Post-8-week Training values significantly higher than the Baseline ($p = 0.000$) and than Post-20-min Running ($p = 0.000$). SHGr presented a significant difference between the Baseline and Post-8-week Training ($p = 0.040$). SFFP in BFGr modified its values with an increasing trend ($p = 0.000$). PP°TFC in BFGr showed a significant decrease ($p = 0.003$) in Post-8-week Training (1.9 kg·cm$^{-2}$) compared to the Baseline (2.4 kg·cm$^{-2}$). In PP°FFP significant decreases were recorded in BFGr and between Post-8-week Training and Baseline ($p = 0.000$), and Post-8-week Training and Post 20 min Running ($p = 0.035$).

**Conclusions:** The adaptation took place after the 8-week training. The adaptations to running barefoot were characterized by causing an increase of the foot's plantar support in TFC and in FFP, as well as a decrease of the plantar pressure peak in both

**PeerJ** _______________________________________________

phases. Also, there is an increased duration of the TFC and FFP, which may be related to an acquired strategy to attenuate the impacts of the ground's reaction forces.

## INTRODUCTION

Currently, endurance running is one of the ways of practicing physical activity that has more adepts in the world, reaching 55 million practitioners only in the United States (_Statista, 2018_). In spite of the health benefits that this type of physical activity gives its practitioners, it is usual for injuries to occur, mainly in the lower limbs (_Hamill et al., 2008_; _Pohl et al., 2008_), affecting between 30% and 75% of the runners every year (_Kluitenberg et al., 2015_). To date it is known that the most common injuries in runners occur in the knee joints (_Taunton et al., 2002_), which in turn are related to the intensity of the ground reaction forces that occur in every stepping cycle (_Daoud et al., 2012_), which can reach 1.5–3 times the runners' body weight (_Lieberman et al., 2010_).

The above has opened the plantar support pattern study focus during endurance running, with several studies stating that when the runners set their forefoot in the first contact with the ground, the recorded GRF values are smaller (_Daoud et al., 2012_). Due to this observation, there has been reflection on methodologies that allow the runners to adopt plantar support with the forefoot. From this, and with an evolutionary approach, a research line was started oriented to the study of running shoeless, or Barefoot Running (BFR). The first studies (_Bramble & Lieberman, 2004_; _Lieberman, 2006_) theorized from an evolutionary approach, stating that the human foot evolved after adopting the bipedal position, achieved by _Australopithecus afarensis_ about 2.2 million years ago (_Susman, 1983_). This led to an interesting debate when it was stated that endurance running was a key factor in the evolution of modern human beings and that during its practice for transportation and hunting reasons there was no mediation of shoes to get the current foot of _Homo sapiens_ (_Bramble & Lieberman, 2004_). In this respect, it is interesting to mention that there are runners who regularly and systematically incorporate that BFR to their training and competitions (_Hollander et al., 2017a_).

Barefoot studies were at first made with an approach aimed at the analysis of kinematic, kinetic variables, and of electric muscular activity that were produced acutely after a barefoot training session (_Franklin et al., 2015_; _Hall et al., 2013_). At present there has been progress toward the incorporation of new variables in the studies (_Alloway et al., 2016_; _Berrones et al., 2016_; _Ekizos, Santuz & Arampatzis, 2017_). However, there is still a lack of longitudinal type studies analyzing the effects that barefoot running produced on variables that affect plantar support (_Hollander et al., 2017b_). Furthermore, there have been few studies on the analysis by support phase in the running technique, with no papers published to date referring to the moment at which the support foot is resisting the whole

body weight over the widest contact surface, which is known as the Flat Foot Phase (FFP) and corresponds to Total Foot Contact (TFC) (*De Cock et al., 2005*).

Because of the above, the objective of this research was to analyze the modifications undergone by TFC and its FFP of young subjects beginning their practice of barefoot running, in its acute and chronic effect (20 min vs. 8 weeks training of BFR).

## MATERIALS AND METHODS

### Participants

The sample consisted of 23 men and 5 women. All subjects in the sample were students of the science of physical activity, aged 20.0 ± 1.5 years, 70.9 ± 10.4 kg body weight, 1.7 ± 0.1 m tall, and a body mass index (BMI) of 24.1 ± 2.5 kg·m$^{-2}$. The subjects were running regularly between 5 and 10 km per week but had no experience in barefoot running. None of the participants had suffered ankle and/or foot injury during the last 6 months before the beginning of this research. The subjects participated voluntarily in this study, signing the informed consent form written and approved according to the guidelines of the Institutional Ethics Committee of the Universidad de Santiago de Chile (Ethics Report No. 184-2018).

It was guaranteed that the subjects complied with the selection criteria through the information gathered from a questionnaire of podalic and postural health and sports history (*Sánchez, Alarcón & Morales, 2017*). On this form, they were also asked which was their dominant foot, defined as that which they use to kick a ball (*Lake, Lauder & Smith, 2011*). Then two groups were formed: Barefoot (BFGr) with $n = 16$ and Shod (SHGr) with $n = 12$, equal in the characterization variables.

### Acute effect protocol

Before making the measurements, every subject rested for 10 min in supine position so as not to alter the foot's postural dynamics (*Jimenez-Ormeño et al., 2011*). Then the body weight and height were measured and the plantar support was evaluated dynamically (Baseline).

Immediately after, each participant ran on a treadmill (Mod. Excite Run 500; Technogym, Cesena, Italy) during 20 min at 3.1 m·s$^{-1}$ with a 5% slope. This speed was chosen because it has been used most often in the studies of running kinematics and evaluation of plantar pressures related to BFR (*Warne et al., 2016*). The subjects in the SHGr ran with their conventional sports shoes, and those of the BFGr ran barefoot.

Immediately after finishing these trials, the subjects were evaluated again to determine the acute effect produced by running barefoot (Post-20-min Running).

### Chronic effect protocol

During the 8 weeks following the Post 20 min Running, the subjects followed a training protocol consisting in the continuous running practice at a self-chosen speed during 24 sessions distributed in 3 weekly sessions. SHGr did it with their conventional sports shoes and BFGr did it completely barefoot.

**Table 1 Determination of the amount of training of the experimental protocol over the 8 weeks and 24 training sessions.**

|  | Week 1 | | | Week 2 | | | Week 3 | | | Week 4 | | |
|---|---|---|---|---|---|---|---|---|---|---|---|---|
| Session no. | 1 | 2 | 3 | 4 | 5 | 6 | 7 | 8 | 9 | 10 | 11 | 12 |
| Minutes | 5 | 7 | 5 | 8 | 10 | 8 | 15 | 10 | 10 | 20 | 10 | 15 |
|  | Week 5 | | | Week 6 | | | Week 7 | | | Week 8 | | |
| Session no. | 13 | 14 | 15 | 16 | 17 | 18 | 19 | 20 | 21 | 22 | 23 | 24 |
| Minutes | 25 | 15 | 25 | 30 | 25 | 30 | 35 | 30 | 35 | 35 | 40 | 35 |

**Figure 1 Temporal scheme of the evaluation and training protocol.** Plant. P° Eval: Plantar pressure evaluation.

Both groups subjects trained together at the same horary and they were permanently controlled by four research helpers, who registered its assistance and the training volume.

To reduce the possibility of the subjects suffering injuries, both groups practiced on a natural grass football field. The amount of training was measured in minutes, and it was increased in a wavelike-manner, as shown in Table 1.

At the end of the protocol, a third evaluation (Post-8-week Training) took place, taking care that the subjects had between 48 and 72 h of rest, aimed at the determination of chronic types of changes. This evaluation was the same as the one carried out in Baseline. The temporal scheme of the evaluations is shown in Fig. 1.

## Data collection and processing

To obtain the plantar print, a Presscam V4 (Sidas©, Voiron, France) baropodoscope with 1,600 receiving sensors was used, and its platform was fixed to the floor, delivering a sequence of images that allowed identifying time, surface, and pressure variables recorded during the plantar support.

The subjects ran on a smooth and horizontal surface 15 m long at a predetermined speed of $3.1 \text{ m·s}^{-1}$. A music guide was used to indicate to the subjects at what time they had to start running and at what time they had to put their foot on the platform. The duration of the music track was 4.8 s, the time needed to cover 15 m at a speed of $3.1 \text{ m·s}^{-1}$. It was considered as a valid attempt only when the three judges gave their approval and five valid attempts were recorded for each foot of the subjects. A valid attempt was considered

when the foot was fully placed on the platform, when the natural running mechanics was not altered (in step frequency and amplitude) and when the support coincides with the time marked in the musical guide. The platform's software was graduated at a receiving frequency of 100 Hz. For the analyses, the average value obtained from the five valid attempts was used.

## Analysis

All the analyses were made taking as reference the time at which a FFP was achieved, a time that is part of the TFC and was characterized by being the moment at which the maximum value of the support surface is achieved (*De Cock et al., 2005*) and coincides with the GRF peak (*Lieberman et al., 2010*). So, among the temporal variables, duration of TFC (DTFC) (ms) and the moment at which FFP is produced were recorded, expressed as the percentage of DTFC (FFP%DTFC). The maximum surface of the TFC (MSTFC) ($cm^2$) and the surface obtained in the FFP (SFFP) ($cm^2$) were also registered. Also, the peak pressure of TFC (PP°TFC) ($kg·cm^{-2}$) and the peak pressure of FFP (PP°FFP) ($kg·cm^{-2}$) were obtained.

From the images obtained from the baropodoscope, the foot strike pattern was identified from the visualization of the foot zone that first made contact with the ground (*Nunns et al., 2013*), yielding three support classifications: forefoot, midfoot, and hindfoot.

## Statistical analysis

Descriptive statistics was used, expressed as mean and standard deviation. For the initial comparison of both groups, for the comparison of the percent change, and for the comparison between the dominant and the nondominant foot, the paired *t*-test was used, and the size of the effect (ES) was calculated by means of Cohen's *d* (ES > 0.8 was considered a strong effect).

To establish effects and interactions that the Time factor produces on each of the variables (Baseline/Post-20-min Running/Post-8-week Training), as well as the effects and interactions that the factor Condition of use of sports shoes produces (SHGr/BFGr), a $3 \times 2$ variance analysis of repeated measures (ANOVA) was made. A post hoc analysis with Bonferroni correction was performed. Homoscedasticity was calculated by means of e Lèvene's test, and sphericity based on Mauchly's test. The size of the effect, expressed as partial eta squared ($\eta_p^2$), indicated the percentage variance in each of the effects and their associated error which is explained by this effect.

Statistical significance was determined as $p \le 0.05$. The analyses were made using the statistical SPSS (V23.0; IBM Inc., Chicago, IL, USA) analysis software.

## RESULTS

Initially, the experiment had 42 subjects, but for the present research, the 28 who completed the experimental protocol were considered. Only two subjects mentioned that they had suffered abrasions of the foot sole's skin, a situation that did no allow them to complete the protocol. The other subjects abandoned the protocol on their own decision.

**Table 2 Characterization variables of the two groups obtained in baseline.**

|  | Shod group | Barefoot group | $p$ | Cohen's $d$ |
|---|---|---|---|---|
| Age (years) | 20.50 ± 1.45 | 19.56 ± 1.41 | 0.10 | 0.66 |
| Weight (kg) | 1.72 ± 0.06 | 1.71 ± 0.08 | 0.12 | 0.14 |
| Height (m) | 74.49 ± 10.72 | 68.18 ± 9.99 | 0.61 | 0.61 |
| BMI (kg·m$^{-2}$) | 25.06 ± 2.55 | 23.33 ± 2.35 | 0.07 | 0.71 |

**Table 3 Results obtained by both groups in each of the evaluation times.**

|  | Shod group | | | Barefoot group | | |
|---|---|---|---|---|---|---|
|  | Baseline | Post-20-Minutes Running | Post-8-Week Training | Baseline | Post-20-Minutes Running | Post-8-Week Training |
| Duration of TFC (ms) | 218 ± 39[a] | 221 ± 38[a] | 228 ± 29 | 168 ± 36[a,b] | 160 ± 41[a,b] | 221 ± 40[b] |
| Moment in which FFP occurs (%) | 48.3 ± 6.2 | 48.8 ± 7.1 | 42.8 ± 6.7 | 46.8 ± 10.9 | 48.3 ± 7.4 | 43.0 ± 7.5 |
| Maximum surface of TFC (cm$^2$) | 104.6 ± 26.0[a,b] | 109.8 ± 28.2[a] | 118.6 ± 25.7[b] | 73.6 ± 18.5[a,b] | 83.8 ± 24.7[a,b] | 111.6 ± 19.8[b] |
| Surface of FFP (cm$^2$) | 82.9 ± 19.8[a] | 81.6 ± 22.1[a] | 87.9 ± 18.7 | 54.8 ± 9.7[a,b] | 60.8 ± 14.6[a,b] | 83.8 ± 15.9[b] |
| Peak Pressure of the TFC (kg·cm$^{-2}$) | 2.0 ± 0.6 | 2.2 ± 0.7 | 2.1 ± 0.4 | 2.4 ± 0.4[b] | 2.1 ± 0.5 | 1.9 ± 0.3[b] |
| Peak pressure of FFP (kg·cm$^{-2}$) | 1.8 ± 0.6 | 1.7 ± 0.6 | 1.6 ± 0.5 | 2.7 ± 0.5[b] | 1.8 ± 0.5[b] | 1.4 ± 0.3[b] |

**Notes:**
[a] Significant differences between groups.
[b] Significant differences between evaluation times.

The two groups did not present significant differences in the characterization variables of age, weight, height and BMI (Table 2).

The comparison between dominant and nondominant foot did not show significant differences between both groups in any of the variables and at any time during the evaluation. That is why in the presentation of the results only the data of the dominant foot were used.

Table 3 shows the values obtained by each of the variables defined in each of the evaluation times in each of the groups.

The shoe wearing condition factor (Shod Group/Barefoot Group (BFGr)) produced significant effects in all the variables. Also, the evaluation time or moment (Baseline/Post-20-min Running/Post-8-week Training) factor produced significant effects in all the variables, except in PP°TFC. The variables that showed interaction between the time and group factors were DTFC, MSTFC, SFFP and PP°TFC.

## Duration of TFC

In the duration of TFC variable, a significant effect of the time factor ($F = 22.566$, $p = 0.000$, $\eta_p^2 = 0.465$, statistical power = 1.000) and of the condition factor ($F = 107.912$, $p = 0.000$, $\eta_p^2 = 0.975$, statistical power = 1.000) was found separately. The value of partial eta squared indicates that the time factor by itself affects only 46.5% of the differences, and the condition factor 97.5%. It had significant interaction of both factors ($F = 12.795$, $p = 0.000$, $\eta_p^2 = 0.330$, statistical power = 0.995). Considering the whole sample, the comparison between pairs obtained from the post hoc analysis expressed significant differences thanks

to the condition factor ($p = 0.005$) and the time factor, expressed in the fact that the Post-8-week Training evaluation was different from the Baseline ($p = 0.000$) and a Post-20-min Running ($p = 0.000$). In the Baseline (SHGr = 218 ms against BFGr = 168 ms, $p = 0.001$) and in the Post-20-min Running (SHGr = 221 ms against BFGr = 160 ms; $p = 0,000$), a significant difference was found between the groups. It is therefore seen that the groups begin unequal, but in the last evaluation they become equal. In SHGr no differences were seen between the evaluation times, but in BFGr the changes were seen because there were significant differences between Baseline and Post-8-week Training ($p = 0.000$), and between Post-20-min Running and Post-8-week Training ($p = 0.000$), tending to increase in time.

## The moment in which FFP is produced

The moment in which FFP is produced expressed as percent of total TFC time was influenced by the time factor in 13.2% of the variance ($F = 3.943$, $p = 0.025$, $\eta_p^2 = 0.132$, statistical power = 0.684). The condition factor had a more important effect, accounting for 99.1% of the variance ($F = 2831.420$, $p = 0.000$, $\eta_p^2 = 0.991$, statistical power = 1.000), but no significant differences were established between groups ($p = 0.724$). In the whole sample, a significant increase ($p = 0.029$) was only found in Post-20-min Running (48.5%) respect to the Baseline (42.9%), indicating that in the acute effect the moment at which FFP occurs is delayed.

## Maximum surface of TFC

Maximum surface of TFC also presented significant effects of the Time factor ($F = 27.224$, $p = 0.000$, $\eta_p^2 = 0.511$, statistical power = 1.000) and of the shod condition ($F = 629.425$, $p = 0.000$, $\eta_p^2 = 0.960$, statistical power = 1.000) separately. The time factor affects significantly in MSTFC, the partial eta squared value indicates that this factor accounts for 51.1% of variance. The condition factor is significant and accounts for 96.0% of the variance. A significant interaction of both factors ($F = 6.131$, $p = 0.006$, $\eta_p^2 = 0.191$, statistical power = 1.000) was seen, its partial eta squared value accounting for 19.1% of the variance. Considering the whole sample, the comparisons by pairs indicated significant differences between groups ($p = 0.013$) and at all evaluation times ($p < 0.05$). This variable increased throughout the training protocol. In the Baseline (SHGr = 104.6 cm$^2$ against BFGr = 73.6 cm$^2$, $p = 0.001$) and in Post-20-min Running (SHGr = 109.8 cm$^2$ against BFGr = 83.8 cm$^2$, $p = 0.015$) there were significant differences between groups, but in the last evaluation both groups presented statistical equality of the variable (SHGr = 118.6 cm$^2$ against BFGr = 111.6 cm$^2$, $p = 0.424$). Within BFGr it was found that the last evaluation presented values significantly greater than the Baseline ($p = 0.000$) and the Post-20-min Running ($p = 0.000$). SHGr presented significant difference between Baseline and Post-8-week Training ($p = 0.040$).

## Surface of FFP

The surface of FFP presented similar results. The time factor presented a significant effect on the variable ($F = 23.897$, $p = 0.000$, $\eta_p^2 = 0.479$, statistical power = 1.000), the same as the

condition factor ($F$ = 718.469, $p$ = 0.000, $\eta_p^2$ = 0.965, statistical power = 1.000). The latter has a greater effect on the variance (96.5%). A significant interaction was seen between both factors ($F$ = 10.622, $p$ = 0.000, $\eta_p^2$ = 0.290, statistic power = 0.985). Considering the whole sample, the analysis of comparisons by pairs showed that SHGr presents greater support surface values than BFGr ($p$ = 0.004). Also, there are significant differences between all the evaluation times ($p$ = 0.000), characterized by an increase of the variable in Post-8-week Training (85.8 cm$^2$) and a decrease in Post-20-min Running (71.2 cm$^2$) with respect to the Baseline (78.9 cm$^2$). In the Baseline and in the Post-20-min Running, significant differences were found between both groups ($p$ < 0.01), with SHGr presenting the largest values. The differences disappear in Post-8-week Running ($p$ = 0.530). In SHGr there were no differences between the evaluation moments, but BFGr modified its values throughout the experiment with a marked increasing trend ($p$ = 0.000).

### Peak plantar pressure of TFC

Peak Plantar Pressure of the TFC was characterized by tolerating significant effects of the condition factor ($F$ = 818.872, $p$ = 0.000, $\eta_p^2$ = 0.969, statistical power = 1.000) and because both factors interacted significantly ($F$ = 3.646, $p$ = 0.033, $\eta_p^2$ = 0.123, statistical power = 0.647). In the analysis by pairs no significant differences could be established between groups ($p$ = 0.792), and only in BFGr a significant decrease ($p$ = 0.003) was verified in Post-8-week Training (1.9 kg·cm$^{-2}$) respect to the Baseline (2.4 kg·cm$^{-2}$).

### Peak plantar pressure of FFP

Peak Plantar Pressure of FFP was influenced significantly by the time factor ($F$ = 8.670, $p$ = 0.001, $\eta_p^2$ = 0.250, statistical power = 0.960), and by the condition factor ($F$ = 486.601, $p$ = 0.000, $\eta_p^2$ = 0.949, statistical power = 1.000) The latter factor influenced in 94.9% of the variance, a value greater than that obtained by the time factor (25.0%). In the whole sample, the pressure values decreased significantly in time, finding a greater difference ($p$ = 0.001) between Post-8-week Training (1.5 kg·cm$^{-2}$) and Baseline (1.9 kg·cm$^{-2}$). Significant differences were recorded only in BFGr between Post-8-week Training and Post-20-min Running ($p$ = 0.035).

### Plantar support pattern

Table 4 shows the distribution of the sample according to the registered plantar support pattern, where it was seen that both groups started with different distributions of the plantar support pattern, marked by a greater percentage of subjects who stepped on their hindfoot in SHGr at all the evaluation times. BFGr shows an increase of the proportion of subjects who stepped on the hindfoot in Post-8-week Training.

Due to the differences found for both groups in the Baseline plantar support, an analysis was made of the percent differences detected at each of the evaluation times with respect to the measurements recorded in the Baseline. The comparisons of the groups can be seen in graphs 1–6, whose results indicate that the differences were greater and significant for almost all the variables in Post-8-week Training, showing DTFC ($p$ = 0.0002, ES = 1.66), MSTFC ($p$ = 0.0003, ES = 1.30), SFFP ($p$ = 0.0001, ES = 1.74), PP°TFC

**Table 4 Distribution of strike pattern in both groups in the three assessment moments.**

|  | Shod group | | Barefoot group | |
|  | n | % | n | % |
| --- | --- | --- | --- | --- |
| Baseline |  |  |  |  |
| Hindfoot | 9 | 75.0 | 6 | 37.5 |
| Midfoot | 1 | 8.3 | 3 | 18.8 |
| Forefoot | 2 | 16.7 | 7 | 43.8 |
| Post-20-min Running |  |  |  |  |
| Hindfoot | 10 | 83.3 | 6 | 37.5 |
| Midfoot | 0 | 0.0 | 5 | 31.3 |
| Forefoot | 2 | 16.7 | 5 | 31.3 |
| Post-8-week Training |  |  |  |  |
| Hindfoot | 8 | 66.7 | 10 | 62.5 |
| Midfoot | 1 | 8.3 | 0 | 0.0 |
| Forefoot | 3 | 25.0 | 6 | 37.5 |

($p = 0.0064$, ES = 1.09) and PP°FFP ($p = 0.0221$, ES = 0.89), with the BFGr always as the group presenting the largest percent changes, which varied between 27.6% and 58.6%. It should be noted that there were no significant differences in any of the Post-20-min Running variables.

## DISCUSSION

The objective of this study was to analyze the modifications undergone by the plantar support in its TFC Phase and its FFP of young subjects who are beginning barefoot practice running, both in its acute and chronic effects after 8 weeks of adaptation. Practically all the dependent variables were found to be influenced by the time factors and the use of shoes, with the latter factor as the one that had the most important influence.

Of the 42 participants who started the study, two could not continue because they suffered skin abrasions during the diagnostic evaluations. These injuries occurred during the diagnostic evaluation due to friction between the treadmill and the foot's sole. According to various studies (*Altman & Davis, 2016*; *Hollander et al., 2017a*; *Hryvniak, Dicharry & Wilder, 2014*), the frequency of injuries in subjects who run barefoot is the same as that in those who wear shoes, but the former tend to suffer more injuries on the foot sole, as happened in the present study. To date there are studies of cases referring to miotendinous injuries (*Cauthon, Langer & Coniglione, 2013*) and bone edema (*Ridge et al., 2013*), but not to skin abrasions. Studies are needed on this issue which has an influence at the beginning of the adaptation to barefoot running.

One aspect to discuss in this paper is the fact that the sample studied had no experience training long distances running, an aspect that could have influenced the effects recorded because of the training protocol. However, it is possible to indicate that the statistical analysis allowed to discriminate against the effects that the protocol produced in the whole sample and the sample divided by groups, thus identifying intra-group and inter-group

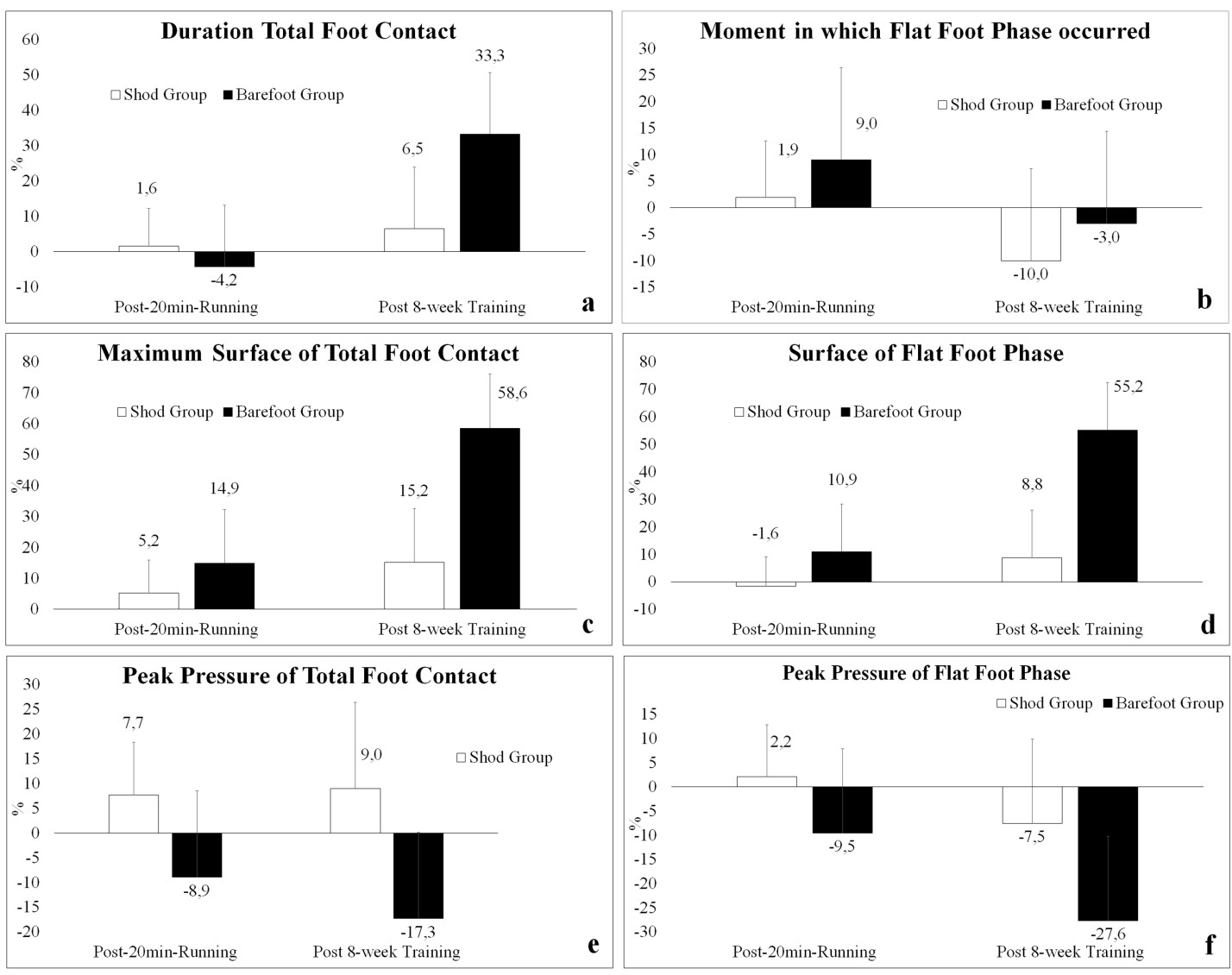

**Figure 2** Graphs of the differences detected at each of the evaluation times compared to the measurements registered in Baseline, expressed as percent. (A) Duration of total foot contact. (B) Moment at which the flat foot phase occurs. (C) Maximum surface of total foot contact. (D) Surface of flat foot phase. (E) Peak pressure of total foot contact. (F) Peak pressure of flat foot phase.

effects. In this regard, the results showed that the significant changes experienced in four of the six variables studied corresponded to the subjects of the group that trained barefoot. Likewise, and according to the graphs shown in Fig. 2, it was possible to determine that the magnitude of the changes was always greater in the group that trained barefoot. This allows us to ponder that if the training effect alone was the main one, the changes would have been the same in both groups, a situation that did not occur.

The decrease of the duration time of TFC has been reported frequently in research dealing with the acute effect of running barefoot (*De Wit, De Clercq & Aerts, 2000*; *Mei et al., 2015*; *Fleming et al., 2015*; *Lieberman et al., 2015*), an aspect that could be verified in this research, although not with significant effects in BFGr. In the long term however,

it was shown that the duration time of total TFC in both groups increased in time, but only in the group that trained barefoot this 33.3% increase was significant after the 8 weeks of training. It should be noted that this result agrees with a recent 12-week experimental study of adaptation (*Muñoz Jiménez et al., 2018*). It seems that long-term adaptations go in the opposite direction as short-term adaptations, and they tend to equal those subjects who wore shoes. However, it should be mentioned that SHGr always showed longer plantar support time values.

The maximum TFC and FFP surface has not been reported as a studied variable in any of the revisions published on this matter (*Hall et al., 2013*; *Hollander et al., 2017b*), but in the present research this variable increased significantly in both groups over time, with a greater change in BFGr, which underwent a 58.6% increase, and 55.2% in TFC and FFP, respectively, compared to 15.2% and 8.8% in TFC and FFP, respectively, of SHGr after the 8-week training. There are two studies that may shed light on the plantar support surface which would have as protagonist the increased cross section of intrinsic foot muscles in subjects running with minimalist footwear (*Miller et al., 2014*; *Johnson et al., 2016*), although this is not necessarily related to modifications of the foot's dimensions (*Sánchez Ramírez & Alegre, 2019*). The use of shoes marked differences between both groups in the first evaluations, but in the last one both groups were statistically equalized. SHGr always showed higher values.

In this research, it was also shown that the pressure peak obtained during TFC and FFP decreased significantly after the 8-week protocol in BFGr, showing a variation of −17.3% and −27.3%, respectively. The increase of the plantar pressure values has been related mainly to the forefoot strike pattern and secondarily with the non-use of shoes or the use of minimalist shoes (*Kernozek et al., 2014*), where it was possible to see a pressure increase in the metatarsal heads (*Szulc et al., 2017*). A longitudinal study looked into this variable after 4 weeks of adaptation to the running with minimalist shoes, finding that the subjects underwent an increase of the maximum support pressure values and an increase of the proportion of subjects who adopt a forefoot strike pattern (*Warne et al., 2014*). In spite of the discordant results, it is interesting to note that the obtained pressure values can be associated with the decrease of the force peak registered in barefoot runners in most studies (*Hall et al., 2013*; *Hollander et al., 2017b*), considering that, due to the characteristics of the equipment used, the maximum value recorded over the whole surface of the foot was obtained. This makes it possible to infer that this decrease could be related to the increases of the recorded plantar support surface, presumably due to a uniform distribution of the foot sole pressures. The latter variable has not been studied intensively, so it is proposed to continue its study.

The time at which FFP occurs showed a significant increase of the acute effect, showing a delay in the appearance of this phase in both groups, in agreement with what was reported in another study in which it was shown that the GRF peak time, which coincides with the FFP moment (*Lieberman et al., 2010*) occurs at 39% of the total duration of the support in barefoot runners and at 41.4% in shod runners running at 3.5 m·s$^{-1}$ (*De Wit, De Clercq & Aerts, 2000*). With this result it is possible to infer that the runner

takes more time in the first foot contact sub-phase as a force absorption strategy (*De Cock et al., 2005*).

The use of sports shoes, due to the additional shock absorption delivered by the foot, makes people support the foot with a greater tendency to a rearfoot strike. When a person is deprived of the shoe, it is probable that, as a shock-absorbing strategy, that person begins striking the forefoot, otherwise, the impact can have a greater effect, causing injuries (*Altman & Davis, 2016*). In this research the plantar support pattern started differently, but it became equal after the 8 weeks of intervention. The BFGr subjects tended to a hindfoot strike and the SHGr subjects almost did not modify their plantar support pattern, results which are quite discordant with the previous studies, which show changes in this variable oriented to forefoot strike. However, it is interesting to indicate that the evidence which supports these statements is still not conclusive (*Hollander et al., 2017b*). This study contributes to other results that require deeper research.

## CONCLUSIONS

The adaptations to barefoot running undergone by plantar support take place after an 8-week training period and not after a 20-minute session. In this respect, the adaptation to running barefoot produces an increase of the surface of plantar support of the foot in the TFC and in the FFP, as well as a decrease of the plantar pressure peak in both phases. Also, an increase in the duration of the TFC and in the FFP is observed, which may be related to an acquired strategy to attenuate the impacts of the reaction forces of the ground.

### Funding
This work was supported by the Dirección de Investigaciones Científicas y Tecnológicas (DICYT) of the Universidad de Santiago de Chile (Project No. 021887SR). The funders had no role in study design, data collection and analysis, decision to publish, or preparation of the manuscript.

### Grant Disclosures
The following grant information was disclosed by the authors:
Universidad de Santiago de Chile: 021887SR.

### Competing Interests
The authors declare that they have no competing interests.

### Author Contributions

- Celso Sánchez-Ramírez conceived and designed the experiments, performed the experiments, analyzed the data, prepared figures and/or tables, authored or reviewed drafts of the paper, and approved the final draft.
- Luis M Alegre conceived and designed the experiments, authored or reviewed drafts of the paper, and approved the final draft.

## Human Ethics

The following information was supplied relating to ethical approvals (i.e., approving body and any reference numbers):

This research was approved by the Institutional Ethics Committee of the University of Santiago de Chile, through an ethical report No. 184 of April 10, 2018.

## Data Availability

The raw measurements are available in the Supplemental Files.

## Supplemental Information

Supplemental information for this article can be found online at http://dx.doi.org/10.7717/peerj.8862#supplemental-information.

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
