# Peer review of "Plantar support adaptations in healthy subjects after eight weeks of barefoot running training"

_PeerJ, doi:10.7717/peerj.8862_

## Round 0.1 · original submission · Minor Revisions

Three reviewers generally provided positive comments about your manuscript. Authors should clarify in the manuscript several methodological aspects following the specific comments of the reviewers.

Reviewer 1 ·

Basic reporting

References
References in the main document should be revised.

Experimental design

Method
The 8-week training should be better explained. For instance: Who watched the runners perform every minute of the training? It has been controlled? Have they been done individually or in groups?

line 205: It is said that the participants have no experience in barefoot running but they appear not to be experienced in running too with only 5 to 10 kms a week. The mentioned adaptations could be part of the changes produced by the adaptation to the race and training, and not to the barefoot running?
line 126: The running speed in the test was 3.1 m·s. It is well known that the speed and fatigue influences the foot strike. If participants are not experienced runners, How does it was taken in account? Not all runners are comfortable with the same running paces or volumes.
line 145: At the end of the 8 weeks protocol, How was the measurement? It is supposed that the participants ran again the 20 minutes in the treadmill as the accutte changes protocol.
line 155: It seems that when the data is taken in 15 meters of stroke with the baropodoscope, the initial running conditions on the treadmill change (f.e. the 5% slope) Could that affect the data?
line 156: “A music guide was used to indicate to the subjects at what time they had to start running and at what time they had to put their feet on the platform”. This part of the protocol should be further explained. What is the role of music exactly?
line 159: “...when the natural running mechanics was not altered (in step frequency and amplitude)”. How has the amplitude and stride frequency been controlled?
line 199: The mortality of the sample is quite high. Why was it?

Validity of the findings

line 205: If there are no significant differences between dominant and non-dominant feet, could it be better to present the data of the average of both feet?

Additional comments

Dear authors, first of all, I congratulate you on the manuscript. I have listed some general comments below for your consideration. Please consider them as constructive recommendations to enrich your study.

Reviewer 2 ·

Basic reporting

no comment

Experimental design

no comment

Validity of the findings

no comment

Additional comments

This manuscript entitled “Plantar support adaptations in healthy subjects after eight weeks of barefoot running training” primarily aimed to analyze the modifications undergone by the Total Foot Contact phase and its Flat Foot Phase in subjects beginning the practice of barefoot running, in its acute and chronic effects. The authors bring an interesting study, but there are still some problems that can not up this review to a publishing level. Some suggestions are listed in the specific comments below.

Specific comments:
1. In the abstract section, line23-24, delete ‘20.0 ± 1.5 years, 70.9 ± 10.4 kg of body weight, 1.7 ± 0.1 m tall, and a BMI of 24.1 ± 2.5 kg•m²’. Line 46, remove ‘and not in an acute manner’.
2. Too many abbreviations in this manuscript, abbr such as ‘duration of TFC (DTFC)’ is unnecessary.
3. In the introduction section, line 59, 30%-75%. Line 81, make revise for ‘of activity electric muscular activity’.
4. Line 82, ‘plantar pattern was modified from the hindfoot to the forefoot’. It should be noticed that the change from the hindfoot to the forefoot is the foot strike pattern, not the plantar pattern.
5. In the methods section, ‘activity, aged 20.0 ± 1.5 years, 70.9 ± 10.4 kg body weight, 1.7 ± 0.1 m tall, and a body mass index (BMI) of 24.1 ± 2.5 kg•m-².’ Is the demographic info for men, women or all subjects, please make it revised.
6. Line 118, 134, acute effect protocol, chronic effect protocol
7. Line 120, 126, ‘for’ instead of ‘during’, with a slope of 5%. Line 131, these trials.
8. Line 137, why all participants chose the self-chosen speed in their training sessions?
9. Line 159, I recommend the authors rewrite ‘from the images obtained of the support’ to make it more understandable.
10. Line 183, which t-test was used for the statistical analysis? Effect size. Line 195 the statistical analysis software SPSS.
11. In the results section, line 199-203, remove it from ‘results’ or it can be briefly described in the methods. Was it only 2 people who suffered the sole abrasions and bruises? As on one have previous experience of barefoot running. Line 205-207, should be written in the methods part.
12. I suggest the authors trim down the results section.
13. Please check the language and grammar mistakes throughout the whole article to improve clarity.
14. The reference format needs some adjustments.
15. Full-long lines is needed for table 1,4.

·

Basic reporting

The article presents a clear experimental approach to understanding the adaptations due to barefoot running in a 8 week period. The specific outcomes and conclusions are clear and relevant to the larger knowledge pool.

While in certain sections of the paper, the English could be improved to keep the flow clear, overall the paper is written and presented well.

Experimental design

The design of the experiment is comprehensive and the statistical analysis is sound.

Validity of the findings

Findings are valid and the statistical analysis carried out is appropriate and detailed.

Additional comments

The language can be improved in certain sections and the meaning of certain phrases can be improved on

Suggested modifications
line 26, at the end 'of' a running

Please clarify the following;
line 38-39, what is meant by "the last evaluation values"
line 77, what "that practice" refers to
line 80, sentence is too long and tends to lose meaning. Simplify sentence
line 88, 'hat' should be' that'
line 97, significance of 8 weeks in the context of this study
line 335-339 Sentence can be written shorter and clearer
Figure 2B Y axis (% label is missing)

---

## Round 0.2 · Minor Revisions

Dear authors, please consider the last comments of the reviewer.

Reviewer 2 ·

Basic reporting

I’m glad receive a revised version of the manuscript. Overall, this manuscript almost shows a fair enough level for publication, but there are still some works that need to do for a better version. Further revisions are needed.

Experimental design

no comment

Validity of the findings

no comment

Additional comments

General comments
1. Line 35-36, ‘a significant increase (p = 0.029)’.
2. Line 42-43, ‘and between Post-8-week Training and…’.
3. Line 44-46, The adaptation took place after the 8-week training. The adaptations to running barefoot were characterized by causing an increase…
4. Line 53-54, Giving the deadline year of the data (55 million practitioners in the United States). ’55 million’ rather than ’55 millions’.
5. Line 56, mainly in the lower limbs.
6. Line 59, remove ‘mainly’.
7. Line 82-83, barefoot running ‘produced’ on variables that affect plantar support.
8. Line 91, using abbr. TFC, FFP.
9. Line 92, ‘in its acute and chronic effect (20 min VS 8 weeks training of BFR)’.
10. Line 101, ‘The subjects were running regularly between 5 and 10 km per week’. Line 132, running at a self-chosen speed.
Why the authors replied that the subjects had no experience as runners?
In this case, why the 20 min acute running did not use the same speed (self-chosen speed).
11. Line 133, delete ‘during eight weeks’, as it is duplication.
12. Line 157-158, line 162-163, change the sentence order as: It was considered as a valid attempt only when the three judges gave their approval and five valid attempts were recorded for each foot of the subjects.
13. Line 188, Showing the classification of ES.
14. Giving the subtitles for results for the clarity purpose.
15. Adding the calculated ES values to the results.

---

## Author Rebuttal · Round 0.2

UNIVERSIDAD
DE SANTIAGO
DE CHILE

Santiago, Chile. January 31st, 2020.

**Dr.**
**Amador García Ramos**
**Academic Editor, PeerJ**
**PRESENT**

Dear Editor

We thank the reviewers for the kindness and time spent preparing their comments and suggestions, which have been of great value to improve our paper. We have considered each of the comments and made the respective modifications.

We believe that with the modifications made, our paper is now able to be published.

Dr. Celso A. Sánchez Ramírez
Assistant Professor
Physical Activity, Sport and Health School

On behalf of all authors.

### *Reviewer 1*

#### *Basic reporting*

*References*
*References in the main document should be revised.*

Have been reviewed all references in the main document and have been corrected those were without reference format or had mistakes.

#### *Experimental design*

*Method*
*The 8-week training should be better explained. For instance: Who watched the runners perform every minute of the training? It has been controlled? Have they been done individually or in groups?*

The following paragraph has been added:
"Both groups subjects trained together at the same horary and they were permanently controlled by four research helpers, who registered its assistance and the training volume."

*line 205: It is said that the participants have no experience in barefoot running but they appear not to be experienced in running too with only 5 to 10 kms a week. The mentioned adaptations could be part of the changes produced by the adaptation to the race and training, and not to the barefoot running?*

The reference corresponds to line 105.

This comment is very interesting and relevant because it criticizes a basic aspect of the investigation. Although it is not included explicitly, the statistical analysis makes it possible to discriminate about the effects that the protocol produced in the whole sample and in the sample separated by groups, identifying intra-group and inter-group effects. In this regard, it is possible to indicate that the results showed differences within the group that trained barefoot in the variables Duration of TFC, Maximum surface of TFC, Surface of FFP, Peak pressure of TFC and Peak pressure of FFP.

These results allow us to ponder that, although the subjects of the study had no experience training with resistance running, the significant changes experienced in four of the six variables studied corresponded to the subjects of the barefoot group. Likewise, and according to the graphs shown in Figure 2, it was possible to determine that the magnitude of the changes was always greater in the group that trained barefoot.

If the training effect alone was the main one, the changes would have been the same in both groups, a situation that did not occur.

These clarifications were included like a paragraph, allowing improved the discussion section quality. I appreciate the observation of the reviewer.

line 126: The running speed in the test was 3.1 m·s. It is well known that the speed and fatigue influences the foot strike. If participants are not experienced runners, How does it was taken in account? Not all runners are comfortable with the same running paces or volumes.

I appreciate this observation since it is likely that the use of a single speed for all subjects could have affected the results of the acute protocol, which consisted of running only once on a treadmill at 3.1 $m·s^{-1}$, and was precisely the moment of evaluation that did not show significant differences within groups. This speed was chosen to partially resemble the methodology used in several other studies (Warne et al., 2014; Fleming et al., 2015; Warne et al., 2016; Fuller et al., 2017).

Regarding the foot strike pattern and the influence that speed and fatigue can exert on this topic, I consider it very true, however, it is also important to indicate that the speed of 3.1 $m·s^{-1}$ was used to obtain of plantar pressures. In this regard, the distance to be covered by each of the participants was 15 m, which is why volume and fatigue are not presented as factors that have an important influence. It also seems pertinent to indicate that the foot strike pattern was only a descriptive variable and that it was not submitted to the 3x2 analysis of variance (ANOVA).

line 145: At the end of the 8 weeks protocol, How was the measurement? It is supposed that the participants ran again the 20 minutes in the treadmill as the accutte changes protocol.

In reality, after 8-weeks, the subjects were evaluated as the Baseline, that is, immediately after the 10-minute rest in the supine position. In that evaluation, the 20-minute run on the treadmill was not done again. I appreciate your question, and thanks to it, a clarification of the chronic effect protocol is included in that paragraph.

line 155: It seems that when the data is taken in 15 meters of stroke with the baropodoscope, the initial running conditions on the treadmill change (f.e. the 5% slope) Could that affect the data?

Yes, the 20-min run conditions on the treadmill might affect the acute effect protocol data. However, both groups carried out the intervention of the protocol and measurement on the baropodoscope under the same conditions, a situation that does not affect the statistical analysis of the effects produced by the barefoot running practice.

[Figure]

line 156: "A music guide was used to indicate to the subjects at what time they had to start running and at what time they had to put their feet on the platform". This part of the protocol should be further explained. What is the role of music exactly?

The usefulness of the music track was explained in the document, adding the following paragraph:

A music guide was used to indicate to the subjects at what time they had to start running and at what time they had to put their foot on the platform. The duration of the music track was 4.8 s, the time needed to cover 15 m at a speed of 3.1 m·s$^{-1}$.

line 159: "...when the natural running mechanics was not altered (in step frequency and amplitude)". How has the amplitude and stride frequency been controlled?

I appreciate this observation. For clarification, it is has included the next paragraph in the document:

"This condition was controlled by three judges trained in this task. Each of them watched the subjects' running and issued his judgment. It was considered as a valid attempt only when the three judges gave their approval".

line 199: The mortality of the sample is quite high. Why was it?

In the first place, it is good to indicate that the subjects participated voluntarily in the study, and there were no resources to ensure the fidelity of the sample. This situation caused several subjects lost training sessions, which caused them to be discarded from the study. Thus, only those subjects who performed the entire training protocol in the times allocated for this were considered for analysis.

The paragraph was changed in the document as follows:

"Only two subjects mentioned that they had suffered abrasions of the foot sole's skin, a situation that did no allow them to complete the protocol. The other subjects abandoned the protocol on their own decision".

Validity of the findings
line 205: If there are no significant differences between dominant and non-dominant feet, could it be better to present the data of the average of both feet?

True, the average of both feet could have been used, but to be able to standardize the results with other similar studies (Nagel et al., 2008; Bergstra et al., 2014; Kernozek et al., 2014; Warne et al., 2014; Barcelona et al., 2017), only the data of the dominant foot was used. It is interesting to consider your observation for future articles. Your feedback is appreciated.

[Figure]

Comments for the author
Dear authors, first of all, I congratulate you on the manuscript. I have listed some general comments below for your consideration. Please consider them as constructive recommendations to enrich your study.

I much appreciate your comments and observations, which have contributed to improving our research article. Have my best regards.

**Reviewer 2**

Basic reporting
no comment

Experimental design
no comment

Validity of the findings
no comment

Comments for the author
This manuscript entitled "Plantar support adaptations in healthy subjects after eight weeks of barefoot running training" primarily aimed to analyze the modifications undergone by the Total Foot Contact phase and its Flat Foot Phase in subjects beginning the practice of barefoot running, in its acute and chronic effects. The authors bring an interesting study, but there are still some problems that can not up this review to a publishing level. Some suggestions are listed in the specific comments below.

Specific comments:

1. In the abstract section, line23-24, delete '20.0 ± 1.5 years, 70.9 ± 10.4 kg of body weight, 1.7 ± 0.1 m tall, and a BMI of 24.1 ± 2.5 kg•m²'. Line 46, remove 'and not in an acute manner'.

The indicated segment was deleted.

2. Too many abbreviations in this manuscript, abbr such as 'duration of TFC (DTFC)' is unecessary.

The abbreviations DTFC and FFP% DTFC were removed, simplifying the summary.

3. In the introduction section, line 59, 30%-75%. Line 81, make revise for 'of activity electric muscular activity'.

The observations were corrected.

4. Line 82, 'plantar pattern was modified from the hindfoot to the forefoot'. It should be noticed that the change from the hindfoot to the forefoot is the foot strike pattern, not the plantar pattern.

The indicated phrase was modified.

5. In the methods section, 'activity, aged 20.0 ± 1.5 years, 70.9 ± 10.4 kg body weight, 1.7 ± 0.1 m tall, and a body mass index (BMI) of 24.1 ± 2.5 kg•m-².' Is the demographic info for men, women or all subjects, please make it revised.

The paragraph was modified to specify the sample:

"The sample consisted of 23 men and 5 women. All subjects in the sample were students of the science of physical activity, aged 20.0 ± 1.5 years, 70.9 ± 10.4 kg body weight, 1.7 ± 0.1 m tall, and a body mass index (BMI) of 24.1 ± 2.5 kg·m-²."

6. Line 118, 134, acute effect protocol, chronic effect protocol

The indicated was modified.

7. Line 120, 126, 'for' instead of 'during', with a slope of 5%. Line 131, these trials.

The indicated was modified.

8. Line 137, why all participants chose the self-chosen speed in their training sessions?

This decision was taken as a research team because the subjects had no experience as runners and even less with barefoot running to avoid injuries in the study participants.

9. Line 159, I recommend the authors rewrite 'from the images obtained of the support' to make it more understandable.

The sentence was rewritten:
"From the images obtained from the baropodoscope, the foot strike pattern was identified…"

10. Line 183, which t-test was used for the statistical analysis? Effect size. Line 195 the statistical analysis software SPSS.

Paired t-test was used. This clarification was included in the document.
The phrase indicated was modified.

11. In the results section, line 199-203, remove it from 'results' or it can be briefly described in the methods. Was it only 2 people who suffered the sole abrasions and bruises? As on one have previous experience of barefoot running. Line 205-207, should be written in the methods part.

The observation is interesting and corresponds to a concern the authors had to write this paper. However, we chose to leave the paragraphs indicated in the results section because they correspond to the report of the analysis of the data that are confronted in the discussion section.

Regarding the subjects who abandoned due to skin abrasions, it is necessary to indicate that only two abandoned for this reason. The subjects participated voluntarily in the study, and there were no resources to ensure the fidelity of the sample. This situation caused several subjects to lose training sessions, which caused them to be discarded from the study. Thus, only those subjects who performed the entire training protocol in the times allocated for this were considered for analysis.

The paragraph was changed in the document as follows:

"Only two subjects mentioned that they had suffered abrasions of the foot sole's skin, a situation that did no allow them to complete the protocol. The other subjects abandoned the protocol on their own decision".

12. I suggest the authors trim down the results section.

I appreciate your suggestion. It would be valuable for us to know what parts of the results section it would be necessary to cut. Please indicate this suggestion in more detail.

13. Please check the language and grammar mistakes throughout the whole article to improve clarity.

The errors indicated have been reviewed and corrected.

14. The reference format needs some adjustments.

Have been reviewed all references in the main document and have been corrected those were without reference format or had mistakes.

15. Full-long lines is needed for table 1,4.

The lines have been added to the tables.

[Figure]

*Reviewer: Pujitha Silva*

Basic reporting

The article presents a clear experimental approach to understanding the adaptations due to barefoot running in a 8 week period. The specific outcomes and conclusions are clear and relevant to the larger knowledge pool.

While in certain sections of the paper, the English could be improved to keep the flow clear, overall the paper is written and presented well.

Experimental design
The design of the experiment is comprehensive and the statistical analysis is sound.

Validity of the findings
Findings are valid and the statistical analysis carried out is appropriate and detailed.

Comments for the author
The language can be improved in certain sections and the meaning of certain phrases can be improved on

Suggested modifications
line 26, at the end 'of' a running

The suggestion was corrected.

Please clarify the following;
line 38-39, what is meant by "the last evaluation values"

Changed for "…showed in Post-8-week Training values…"

line 77, what "that practice" refers to

The activity (BFR) was specified:
"…incorporate that BFR to their training and competitions…"

line 80, sentence is too long and tends to lose meaning. Simplify sentence

The sentence was shortened:
Barefoot studies were at first made with an approach aimed at the analysis of kinematic, kinetic variables, and of electric muscular activity that were produced acutely after a barefoot training session.

line 88, 'hat' should be' that'

The suggestion was corrected.

[Figure]

line 97, significance of 8 weeks in the context of this study

It was specified by incorporating "BFR" into the sentence.

line 335-339 Sentence can be written shorter and clearer

The sentence was simplified.

Figure 2B Y axis (% label is missing)

The figure was modified.

---

## Round 0.3 · accepted · Accept

Congratulations for meeting the publication standards of PeerJ

---

## Author Rebuttal · Round 0.3

UNIVERSIDAD
DE SANTIAGO
DE CHILE

Santiago, Chile. February 27th, 2020.

**Dr.**
**Amador García Ramos**
**Academic Editor, PeerJ**
**PRESENT**

Dear Editor

We thank the reviewers for the kindness and time spent preparing their new comments and suggestions, which have been of great value to improve our paper. We have considered each of the comments and made the respective modifications.

We believe that with the modifications made, our paper is now able to be published.

Dr. Celso A. Sánchez Ramírez
Assistant Professor
Physical Activity, Sport and Health School

On behalf of all authors.

[Figure]

*Reviewer 2*
*Comments for the author:*

*1. Line 35-36, 'a significant increase (p = 0.029)'.*

The word indicated was included.

*2. Line 42-43, 'and between Post-8-week Training and…'.*

The word indicated was included.

*3. Line 44-46, The adaptation took place after the 8-week training. The adaptations to running barefoot were characterized by causing an increase…*

The sentence was modified according to indicated.

*4. Line 53-54, Giving the deadline year of the data (55 million practitioners in the United States). '55 million' rather than '55 millions'.*

The word was modified according to indicated.

*5. Line 56, mainly in the lower limbs.*

The word suggested was replaced.

*6. Line 59, remove 'mainly'.*

The word was removed.

*7. Line 82-83, barefoot running 'produced' on variables that affect plantar support.*

The word was modified according to indicated.

*8. Line 91, using abbr. TFC, FFP.*

The words were replaced by the respective abbreviations.

*9. Line 92, 'in its acute and chronic effect (20 min VS 8 weeks training of BFR)'.*

The sentence was modified according to indicated.

*10. Line 101, 'The subjects were running regularly between 5 and 10 km per week'. Line 132, running at a self-chosen speed.*
*Why the authors replied that the subjects had no experience as runners?*
*In this case, why the 20 min acute running did not use the same speed (self-chosen speed).*

Yes, it seems that the sentence does not explain in the right way the characteristics of the sample regarding its training history. The sentence was changed, as follows:

*"The subjects were running regularly between 5 and 10 km per week but had no experience in barefoot running."*

According to Line 132, I appreciate this observation since it is likely that the use of a single speed for all subjects could have affected the results of the acute protocol, which consisted of running only once on a treadmill at 3.1 m·s-1, and was precisely the moment of evaluation that did not show significant differences within groups. This speed was chosen to partially resemble the methodology used in several other studies (Warne et al., 2014; Fleming et al., 2015; Warne et al., 2016; Fuller et al., 2017).

*11. Line 133, delete 'during eight weeks', as it is duplication.*

The phrase was deleted.

*12. Line 157-158, line 162-163, change the sentence order as: It was considered as a valid attempt only when the three judges gave their approval and five valid attempts were recorded for each foot of the subjects.*

The paragraph was changed according to suggested.

*13. Line 188, Showing the classification of ES.*

The next phrase was included:
*"(ES > 0.8 was considered a strong effect)."*

*14. Giving the subtitles for results for the clarity purpose.*

Subtitles were incorporated into the results.

*15. Adding the calculated ES values to the results.*

The cohen´s d value was included in table 2.
The effect size based on $\eta^2_p$ is shown in the results paragraphs.